# The Orophilous Shrubby Vegetation of the *Juniperetalia hemisphaericae* Order in Sicily: A Refuge Habitat for Many Endemic Vascular Species

**DOI:** 10.3390/plants13030423

**Published:** 2024-01-31

**Authors:** Saverio Sciandrello, Gianpietro Giusso del Galdo

**Affiliations:** Department of Biological, Geological and Environmental Sciences, University of Catania, Via A. Longo 19, 95125 Catania, Italy; g.giusso@unict.it

**Keywords:** conservation, endemic species, *Juniperus*, high-mountain vegetation, phytosociology, habitat, syntaxonomy

## Abstract

An in-depth analysis of the orophilous vegetation dominated by *Juniperus hemisphaerica* in Sicily, based on literature data and unpublished relevés, is presented. A total of 156 phytosociological relevés were processed and analyzed using classification and ordination methods. Overall, 151 vascular plant species were recorded, of which 38% were endemic species, with a dominant hemicryptophytic biological form (49%). Classification of the relevés, supported by ordination, showed two main vegetation groups: one including the *Juniperus* communities of Mt. Madonie (*Cerastio tomentosi*–*Juniperetum hemisphaericae* and *Pruno cupanianae*–*Juniperetum hemisphaericae*), and the other including the vegetation of Mt. Etna (*Bellardiochloo aetnensis*–*Juniperetum hemisphaericae* and *Roso siculae*–*Juniperetum hemisphaericae* ass. nova). Furthermore, a new alliance, *Berberido aetnensis*–*Juniperion hemisphericae*, is proposed for Sicily.

## 1. Introduction

In the frame of phytosociological surveys on the orophilous vegetation of Sicily, the juniper scrub communities belonging to the *Juniperetalia hemisphaericae* Rivas-Martínez & J.A. Molina 1998 are examined. This order, of the *Junipero-Pinetea sylvestris* class, described by Rivas-Martínez & J.A. Molina 1998 in Rivas-Martìnez et al. [1], includes the oro-Mediterranean, upper supra-Mediterranean, and upper supra-oro-sub-Mediterranean climactical, permanent, relict, or secondary dwarf scrublands, growing on the high mountains of the Iberian Peninsula, south Alps, Apennines, Thyrrenic Islands and humid Mauritanian High Atlas (Typus: *Genisto versicoloris*–*Juniperion hemisphaericae* Rivas-Martínez & Molina 1998). The *Juniperetalia hemisphaericae* order includes three alliances: 1. *Cytision oromediterranei* Tx. in Tx. et Oberd. 1958 corr. Rivas-Mart. 1987, which groups the silicicolous orotemperate, sub-Mediterranean dry juniper scrub of the Central Iberian and Cantabrian mountains; 2. *Genisto versicoloris*–*Juniperion hemisphericae* Rivas-Mart. et J.A.Molina in Rivas-Mart. et al., 1999, which groups the silicicolous oro-Mediterranean dry juniper scrub of the Sierra Nevada (Southern Iberian Peninsula); 3. *Pruno prostratae*–*Juniperion sabinae* Rivas-Mart. et J.A.Molina in Rivas-Mart. et al., 1999, which includes the calcicolous supra-oro-Mediterranean and supra-orotemperate sub-Mediterranean dry juniper scrub of the Central Iberian and Cantabrian Mountains. The first phytosociological data on the *Juniperus hemisphaerica* community in Sicily, and in particular on Etna, are to be attributed to Poli [2], who does not highlight the autonomy of this shrubby vegetation, including them in the *Astragalus siculus* vegetation (*Rumici-Astragalion*). Even before that, Frei [3] highlighted the occurrence of extensive *Berberis aetnensis* and *Juniperus hemisphaerica* communities on Mt. Etna, without reporting any phytosociological relevés.

Other authors highlighted the presence on Etna of a subalpine belt (altitudinal range between 1900 m and 2400 m slm) dominated by *Juniperus*, *Berberis,* and *Astragalus* [4,5,6,7,8,9,10,11]. Subsequently, Pignatti et al. [12] described the *Cerastio tomentosi*–*Juniperetum hemisphaericae* for Mt. Madonie, including this association in the alliance *Cerastio*–*Astragalion nebrodensis* (*Erinacetalia* Quezel 1953), which groups together the basiphilous pulvinate vegetation of the Nebrodi and Madonie Mountains. Afterwards, Brullo [13], considering the noteworthy floristic autonomy of the pulvinate vegetation on Sicilian high mountains, proposed to include this vegetation in the order *Erysimo-Jurinetalia bocconei* (*Cerastio-Carlinetea nebrodensis*). Within this order, Brullo [13] included the alliance described by Pignatti et al. [12] and proposed a new alliance, *Armerion nebrodensis*, grouping the acidophilous communities occurring on the same mountains. The *Juniperus hemisphaerica* vegetation occurring on quartzitic soils was referred to the association proposed by Pignatti et al. [12], while the community occurring on basic substrata was considered by Brullo [13] as *Lino-Seslerietum nitidae* subass. *juniperetosum*. Only later, Brullo and Siracusa in Brullo et al. [14] described a new association *Bellardiochloo aetnensis*–*Juniperetum hemisphaericae* for Mt. Etna and proposed to include the high-mountain *Juniperus* communities in a new suballiance/alliance (*Pinenion calabricae*, *Berberidion aetnensis*) of the *Pino-Juniperetea* class. More recently, Raimondo et al. [15] described a new association *Pruno cupanianae*–*Juniperetum hemisphaericae* for Mt. Madonie, including it in the *Berberidion aetnensis* alliance (*Juniperetalia hemisphaericae* order).

The *Berberidion aetnensis* alliance proposed by Brullo et al. [14] groups coniferous forests (*Junipero hemisphaericae*–*Abietetum nebrodensis*, *Junipero hemisphaericae*–*Pinetum calabricae*, *Junipero hemisphaericae*–*Abietetum apenninae*) or dwarf shrublands (*Cerastio tomentosi*–*Juniperetum hemisphaericae*, *Bellardiochloo aetnensis*–*Juniperetum hemisphaericae*) occurring within the supra-oro-Mediterranean bioclimatic belts, with central Mediterranean distribution (Sardinia, Corsica, Sicily, southern Italy). The authors indicate as an association the type *Junipero nanae*–*Pinetum laricionis* [14]). According to Mucina and Theurillat in Mucina et al. [16], since the type of vegetation (*Junipero nanae*–*Pinetum laricionis*) is tree-dominated, although the *Berberidion aetnensis* comprises both coniferous forests and dwarf shrublands, the name is to be considered illegitimate (ICPN art. 29b). Therefore, Mucina and Theurillat in Mucina et al. [16] propose a new name, *Berberido aetnensis-Pinion laricionis*, including it in a new order named *Berberido creticae*–*Juniperetalia excelsae*, which groups the relict sub-Mediterranean and supra-Mediterranean dry pine forests and juniper woods of the Central and Eastern Mediterranean. In addition to the syntaxonomic aspect, this type of high-mountain vegetation hosts floristic elements of particular phytogeographical value worthy of study [17].

The aim of this paper is to provide an updated syntaxonomical classification (at the level of associations and alliances) of the orophilous vegetation dominated by *Juniperus hemisphaerica* in Sicily, taking into account, for each plant community, the diagnostic species, floristic composition, structure, distribution, and ecological features. In addition, an updated inventory of the endemic flora occurring in the juniper scrub on the high mountains of Sicily is provided, with the hope that this should be a starting point for further research and conservation projects on flora threatened by climate change.

## 2. Results and Discussion

### 2.1. Vegetation Analysis

Classification of the relevés, supported by ordination, showed two main vegetation groups (Figure 1): the first one included the *Juniperus* communities of Mt. Etna (cluster A) and the other the vegetation of Mt. Madonie (cluster B). On the Etna volcano, three communities were identified, with different ecological and dynamic features: the first, *Bellardiochloo aetnensis*–*Juniperetum hemisphaericae* (cluster A2), represents a secondary serial community of the *Junipero hemisphaericae*–*Pinetum calabricae*; the second community, described as a new association *Roso siculae*–*Juniperetum hemisphaericae* (cluster A1), represents a serial primary edapho-xerophilous vegetation of high mountains (permaseries) [18]; the last community, *Junipero hemisphaericae-Pinetum calabricae* (cluster A3), represents the forest vegetation (edaphoxerophilous series) growing on more mature volcanic soils with rocky outcrops, within the supra-Mediterranean belt. The scrub vegetation of Madonie refers to *Cerastio tomentosi*–*Juniperetum hemisphaericae* including *Junipero hemisphaericae*–*Abietetum nebrodensis* (cluster B4), which represent the first serial community of *Junipero hemisphaericae*–*Abietetum nebrodensis*, and *Pruno cupanianae*–*Juniperetum hemisphaericae*, including *Lino punctati*–*Seslieretum siculae subass. juniperetosum* (cluster B5), a serial community of mesophilous holm oak woods of *Acero-Quercetum ilicis*. The DCA (Figure 2) shows a clear split into two groups, mainly linked to the different nature of the substrates in the two mountain massifs: carbonatic on Mt. Madonie and volcanic on Mt. Etna. This difference in substrates and age also corresponds to a different floristic richness between the two mountains, clearly higher in the ancient carbonate substrates of Madonie. On the positive side of axis 1 of the diagram are shown the plant communities of Mt. Madonie, with a marginal overlap between the *Cerastio tomentosi*–*Juniperetum hemisphaericae* and *Pruno cupanianae*–*Juniperetum hemisphaerica*, as shown in the cluster analysis, while on the negative side of axis 1 are distributed the communities of Mt. Etna, with a clear separation, in relation to the altitudinal range, between *Bellardiochloo aetnensis*–*Juniperetum hemisphaericae* and *Roso siculae*–*Juniperetum hemisphaericae*. On the negative section of axis 1, we also find the *Junipero hemisphaericae*–*Pinetum calabricae*, which differs from the shrub *Juniperus* communities due to its structure and floristic composition. All this confirms that this latest new association differs floristically for some rosaceae (*Rosa* sp pl.) characterizing the structure of this new community, above all due to its serial primary edapho-xerophilous role of high mountains (1900–2300 m a.s.l.).

The nomenclature, floristic composition, ecology, syndynamic relationships, and chorology of each examined plant community are critically described below:

*Junipero sabinae*–*Pinetea sylvestris* Rivas-Martinez 1965 nom. inv. propos. Rivas-Martínez et al., 2002

Relict supra-Mediterranean and sub-Mediterranean orotemperate dry pine and juniper woods of the Iberian Peninsula

Characteristic species: *Avenella flexuosa* subsp. *iberica*, *Juniperus sabina*, *Orchis spitzelii*

*Juniperetalia hemisphaericae* Rivas-Mart. et J.A. Molina in Rivas-Mart. et al., 1999

Relict sub-Mediterranean and supra-Mediterranean dry scrub of Western Mediterranean

Characteristic species: *Juniperus hemisphaerica*

*Berberido aetnensis*–*Juniperion hemisphericae* Giusso & Sciandrello 2024 (this paper)

Holotypus: *Roso siculae-Juniperetum hemisphaericae* Giusso & Sciandrello 2024 (this paper)

Characteristic species: *Berberis aetnensis* C. Presl, *Rosa sicula* Tratt., *Rosa heckeliana* Tratt.

Description: Supra-oro-Mediterranean permanent or secondary dry juniper scrub growing in the high mountains of Sicily (Mt. Etna and Mt. Madonie). The shrubby structure is dominated by *Juniperus hemisphaerica* C. Presl and *Berberis aetnensis* C. Presl, together with *Rosa sicula* Tratt., *Rosa heckeliana* Tratt., *Prunus cupaniana* Guss., and *Sorbus graeca* (Spach) Lodd. ex S. Schauer. They find their optimum in the oro-Mediterranean belt of Mt. Etna, mainly on silicicolous substrata in rocky environments affected by the prolonged permanence of snow.

#### 2.1.1. *Cerastio tomentosi*–*Juniperetum hemisphaericae* Pignatti and Nimis in Pignatti et al., 1980

Synonyms: *Lino punctati*–*Seslieretum siculae* subass. *juniperetosum* Brullo 1984

Holotypus: rel. 82, Table 9, Pignatti et al. [12].

Diagnostic species: *Allium nebrodense* Guss., *Cerastium tomentosum* L.

Structure and ecology: This shrubland is characterized by a monostratified structure dominated by prostrate shrubs of *Juniperus hemisphaerica* C. Presl, which are associated with a few other shrub species, such as *Rosa sicula* Tratt., *Daphne oleoides* Schreb., *Berberis aetnensis* C. Presl., *Genista cupanii* Guss., and *Astragalus nebrodensis* (Guss.) Strobl. The herbaceous layer includes many species of *Rumici-Astragaleta*, such as *Festuca circummediterranea* Patzke, *Koeleria splendens* C. Presl, *Sesleria nitida* Ten. subsp. *sicula* Brullo & Giusso, *Silene italica* (L.) Pers. subsp. *sicula* (Ucria) Jeanm., *Carlina nebrodensis* Guss. ex DC., *Allium nebrodense* Guss., *Cerastium tomentosum* L., *Petrorhagia saxifraga* (L.) Link subsp. *gasparrinii* (Guss.) Pignatti ex Greuter & Burdet, etc. This vegetation grows on sunny and windy stands, between 1300 and 1900 m, on limestone, dolomite, and quartzite outcrops with shallow and sparsely evolved soils. The juniper scrubs often grow together with the orophilous pulvinate *Astragalus nedrodensis* vegetation and the orophylous *Sesleria nitida* grasslands, forming a mosaic of high-mountain communities. On Madonie, the juniper occupies two facies (calcicole and silicicole), characterized by *Sesleria nitida* subsp. *sicula* and *Genista cupanii*, respectively [14]. On Monte Quacella’s breccias, according to Brullo [13], it assumes a pioneering role and characterizes the *Lino punctati*–*Seslieretum siculae* subass. *juniperetosum*, while on the quartz arenites it constitutes a well-differentiated association from an ecological and dynamic point of view, an acidophilic community linked to cacuminal stations, more or less inclined, but less ventilated and subject to intense erosion, and represents the passage towards the beech forest [13].

Distribution: Madonie Mountains [12].

Dynamic contacts: *Luzulo siculae*–*Fagetum sylvaticae* Brullo, Guarino, Minissale, Siracusa, and Spampinato 1999

Catenal contacts: This edapho-xerophilous community has catenal contacts with the orophilous pulvinate vegetation of *Rumici-Astragaletea* (*Lino punctati*–*Seslieretum siculae* Pignatti & Nimis in Pignatti et al., 1980 corr. [19]; *Astragaletum nebrodensis* Pignatti & Nimis in Pignatti et al., 1980) and with the basiphilous beech forests of *Geranio versicoloris*–*Fagion sylvaticae* Gentile 1970, nom. mut. prop. by [20].

Notes: The phytosociological relevés (n.4) of *Junipero hemisphaericae*–*Abietetum nebrodensis*, woodland vegetation described by Brullo et al. [14] for the Madonie Mountains, are included in cluster 4 (*Cerastio tomentosi*–*Juniperetum hemisphaericae*) due to low coverage (value 1) of *Abies nebodrensis* and high coverage of *Juniperus hemisphaerica* (value 3). In our opinion, the validity of this relict shrub-dominated association should be reviewed.

#### 2.1.2. *Bellardiochloa aetnensis*–*Juniperetum hemisphaericae* Brullo & Siracusa in Brullo et al., 2001

Holotypus: rel. 7, Table 4, Brullo et al. [14].

Diagnostic species: *Bellardiochloa variegata* (Lam.) Kerguélen subsp. *aetnensis* (C. Presl) Giardina & Raimondo

Structure and ecology: Shrubby vegetation colonizing the poorly evolved volcanic soils between 1400 and 1800 m a.s.l., this community occurs in the supra-Mediterranean belt of the Etna mountain. Its shrub structure is given by *Juniperus hemisphaerica* C. Presl and *Berberis aetnensis* C. Presl. The herbaceous layer includes several species of *Rumici-Astragaleta*, such as *Bellardiochloa variegata* (Lam.) Kerguélen subsp. *aetnensis* (C. Presl) Giardina & Raimondo, *Festuca circummediterranea* Patzke, *Astragalus siculus* Biv., *Silene italica* (L.) Pers. subsp. *sicula* (Ucria) Jeanm., *Galium aetnicum* Biv., *Carlina nebrodensis* Guss. ex DC., Viola *aethnensis* (Ging. & DC.) Strobl subsp. *aethnensis*, *Erysimum etnense* Jord., etc. It represents a geographical vicariant of *Cerastio tomentosi*–*Juniperetum hemisphaericae* [14].

Distribution: Etna: Serra la Nave, Mt. Vettore, Piano Provenzana [14], above Mt. Pecoraro, above Mt Conca (P. Provenzano), above Monte Vettore, between Mt. Nero degli Zappini and Mt. Castellazzo, between Mt. Conca and Mt. Nero delle Concazze.

Dynamic contacts: In the more evolved substrates it tends towards the orophilous pine forests *Junipero hemisphaericae*–*Pinetum calabricae*.

Catenal contacts: *Bellardiochloa aetnensis*–*Juniperetum hemisphaericae* comes in contact with orophilous communities, represented by *Astragaletum siculi* or *Phleo ambigui-Secaletum stricti* and by the orophilous forests of *Junipero hemisphaericae*–*Pinetum calabricae*.

#### 2.1.3. *Pruno cupanianae*–*Juniperetum hemisphaericae* Raimondo, Marino & Schicchi 2010

Holotypus: rel. 3, Table 2, Raimondo et al. [15].

Diagnostic species: *Daphne oleoides* Schreb., *Juniperus hemispaherica* C. Presl, *Prunus cupaniana* Guss., *Scutellaria rubicunda* Hornem. subsp. *linneana* (Cornel) Reichinger f.

Structure and ecology: This vegetation grows on more stabilized screes with dolomitic substrata, from 1300 to 1450 m a.s.l. Even this community is dominated by *Prunus cupaniana* Guss., growing together with *Daphne oleoides* Schreb., *Scutellaria rubicunda* Hornem. subsp. *linneana* (Cornel) Reichinger f. and some species of a higher rank, among them Rosa sicula Tratt., *Berberis aetnensis* C. Presl, *Sorbus graeca* (Spach) Lodd. ex S. Schauer, *Amelanchier ovalis* Medik. subsp. *embergeri* Favarger & Stearn, and *Rubus canescens* DC. Raimondo et al. (2010) report high coverage of *Juniperus hemisphaerica* C. Presl in the *Berberidion aetnensis* alliance (*Juniperetalia hemisphaericae* order).

Distribution: Madonie Mountains, on the northern slopes of Monte Quacella (1869 m s.l.m.) [15].

Dynamic contacts: *Aceri campestri*–*Qurcetum ilicis* Brullo 1984.

Catenal contacts: According to Raimondo et al. [15], this association is in catenal contact with the basiphilous and mesophilous series of beech (*Luzulo siculae*–*Fagetum sylvaticae* Brullo, Guarino, Minissale, Siracusa & Spamp. 1999), with the orophilous basiphilous series of holm oak (*Aceri campestri*–*Qurcetum ilicis* Brullo 1984), and with the orophilous vegetation of the *Junipero hemisphaericae*–*Abietetum nebrodensis* Brullo & Giusso in Brullo et al., 2001.

#### 2.1.4. *Roso siculae*–*Juniperetum hemisphaericae* Giusso & Sciandrello 2024 (This Paper)

Holotypus: rel. 27, Table A1

Diagnostic species: *Rosa sicula* Tratt., *Hieracium pallidum* Biv. subsp. *aetnense* Gottschl., Raimondo & Di Grist.

Structure and ecology: Relict dry juniper scrub of Mt. Etna, linked to volcanic substrata with initial soils characterized by rocky outcrops. This plant community is widespread in the oro-Mediterranean belt at an altitude between 1900 and 2300 m a.s.l. (Figure 3). The shrubby structure is dominated mainly by *Juniperus hemisphaerica*, *Berberis aetnensis* C. Presl, *Rosa sicula* Tratt., and *Rosa heckeliana* Tratt., and sporadically by *Rubus aetneus* Tornab. and *Sorbus graeca* (Spach) Lodd. ex S. Schauer, while the herbaceous layer includes species of *Rumici-Astragaleta*, such as *Festuca circummediterranea* Patzke, *Silene italica* (L.) Pers. subsp. *sicula* (Ucria) Jeanm., *Astragalus siculus* Biv., *Bellardiochloa variegata* subsp. *aetnensis*, *Phleum hirsutum* Honck. subsp. *ambiguum* (Ten.) Cif. & Giacom., *Galium aetnicum* Biv., *Calamagrostis epigejos* (L.) Roth, and *Viola aethnensis* (Ging. & DC.) Strobl subsp. *aethnensis*. Due to its ecological peculiarities, *Rosa sicula* Tratt. and *Hieracium pallidum* Biv. subsp. *aetnense* Gottschl., Raimondo & Di Grist. are proposed as the characteristic species of this new association (Holotypus: rel. 27, Table A1). The last one is an exclusive endemic species of Etna described by Gottschlich et al. [21] for Monte Pomiciaro (Etna), among the clearings of a *Genista etnensis* (Raf.) DC. scrubland, at an altitude of 1603 m a.s.l. The growth site (locus classicus) of *H. pallidum* Biv. subsp. *aetnense* indicated by Gottschlich et al. [21] represents probably the secondary habitat of the species, as the primary habitat is represented by the high-mountain juniper scrublands of Etna.

Distribution: Etna: above the beech forest of Timparossa and Mt. Nero (N), above Mt. Conca (Piano Provenzano, E), Mt. Guardinazzi and Mt. Conca (NO), Mt. Palestra and Mt. Pecoraro (O), between Mt. Scavo and Mt. Frumento Supino (O, SO), and Mt. Nero degli Zappini (SO). Furthermore, small nuclei of juniper (not mapped) have been identified on the north and north-east sides of Mt. Frumento delle Concazze and between Serra Perciata and Serra dell’Acqua (Valle del Bove).

Dynamic contacts: Relict primary orophilous scrub vegetation.

Catenal contacts: Generally, at lower altitudes, this association comes in contact with the mesophilous woods of *Epipactito-Fagetum sylvaticae* (between 1500 m and 2000 m s.l.m.) or, in more xeric conditions, with pine woods, whereas at higher altitudes it is in contact and interspersed with *Astragaletum siculi* (Frei 1940) Gilli 1943 corr. [19] or *Festuco circummediterraneae*–*Bellardiochloetum aetnensis* Frei 1940 corr. [19]. On the north side of Mt. Frumento delle Concazze at 1940 m (at lower altitudes), small patches of juniper can be found in contact with the *Betula etnensis* forest (*Cephalanthero longifoliae*–*Betuletum aetnensis* Brullo & Siracusa 2012).

Notes: In the diagnosis, Rivas-Martínez & J.A. Molina 1998 in Rivas-Martìnez et al. [1] clearly indicate the structure of the community type (dwarf scrublands) and the geographical distribution (high mountains of the Iberian Peninsula, southern Alps, Apennines, Thyrrenic Islands and humid Mauritanian High Atlas) of the *Juniperetalia hemisphaericae* order. Subsequently, Mucina & Theurillat in Mucina et al. [16] describe a new order, *Berberido creticae*–*Juniperatalia excelsa*, which includes both dry pine forests and juniper woods, with a geographical distribution in the Central and Eastern Mediterranean. The same authors describe a new alliance, *Berberido aetnensis*–*Pinion laricionis*, which replaces *Berberidion aetnensis*, name illegitimate (ICPN art. 29), described by Brullo et al. [14]. For these reasons, we propose a new alliance be included in the *Juniperetalia hemisphaericae* order, which better characterizes the *Juniperus* communities of high mountains in Sicily from a chorological and structural point of view.

### 2.2. Floristic Remarks on Juniper Vegetation

A total of 151 vascular plant species belonging to 42 families were recorded in the high-mountain juniper communities of Sicily. The most represented families are Asteraceae (17%), Poaceae (13%), and Rosaceae (10%). The life form spectrum indicates the prevalence of hemicryptophytes (49%), chamephytes (20%), and phanero/nanophanerophytes (21%). The endemic flora accounts for 57 taxa (38%): 31 endemic of Sicily and 26 endemic of Italy. The endemic species are mainly localized within the oro-Mediterranean bioclimatic belts (1550–2400 m a.s.l.). In the Madonie Mountains at altitudes between 1300 and 1900 m (36 rels), a total of 116 species were recorded, dominated by the hemicryptophytic biological form (47%), with endemism of around 34%. On Mount Etna, at altitudes between 1400 and 2200 m (120 rels), a total of 58 species were recorded, dominated by the hemicryptophytic life form (52%), with endemism of around 43%. The most important (from a phytogeographic point of view) vicariant endemic species, widespread among the high-mountain juniper scrubs of the two mountain massifs (Madonie and Etna), are Astragalus nebrodensis (Guss.) Strobl. vs. Astragalus siculus Biv., Erysimum bonannianum C.Presl vs. Erysimum etnense Jord., Bellardiochloa variegata (Lam.) Kerguélen subsp. nebrodensis (Asch. & Graebn.) C.Brullo, Brullo, Giusso & Sciandr. vs. Bellardiochloa variegata (Lam.) Kerguélen subsp. aetnensis (C.Presl) Giardina & Raimondo, Centaurea parlatoris Heldr. vs. Centaurea giardinae Raimondo & Spadaro, and Hieracium pallidum Biv. subsp. pallidum vs. Hieracium pallidum Biv. subsp. aetnense Gottschl., Raimondo & Di Grist. The likely explanation for the extremely high endemism in the high mountains of Sicily can be found in the insularity [17] and differentiation of the mountain flora. Therefore, the high-mountain Mediterranean vegetation with shrubs (thorny) can be considered one of the centers of the greatest concentration of endemism in the Mediterranean Basin [12].

### 2.3. Distribution and Conservation

The cartographic analysis, performed with Quantum GIS software, shows the *Juniperus* scrub distribution on Mt. Etna, with 89 polygons and 220 geolocalized points, for a total surface area of 323 ha (Figure 4). According to the reference grid (2 × 2 km), this habitat is currently recorded in 24 cells corresponding to a total surface area of 96 km^2^. The slopes of the volcano most affected by juniper groves are mainly to the north and west, at altitudes between 1900 and 2300 m. The juniper undergrowth of the *Pinus laricio* (not mapped) extends to lower altitudes, up to 1700–1600 m and, in some cases, even to 1500–1400 m, among the *Quercus congesta* oak woods, near Mt. Maletto (Etna NW), characterizing the shrubby structure of the forest community with high cover values. In addition, it is possible to find isolated junipers, even among the oak woods of *Quercis ilex*, near Monte Tre Frati (Etna W), and among the oak woods of *Quercus cerris*, near C.da Giarrita (Etna E). Etna, due to its high altitudes, is the only one of the Sicilian mountain systems to host the relict primary orophilous vegetation dominated by *Juniperus hemisphaerica*, between 1900 and 2300 m a.s.l., in contact (top) with the vegetation of *Astragalus siculus*, which rises to almost 2500 m, and (below) with beech woods (between 1800 and 2000 m). On Etna, the *Juniperus hemisphaerica* vegetation plays two different ecological and dynamic roles: at low altitudes, it occupies the acidophilic climatic series of the beech (*Epipactido*-*Fageto sylvaticae sigmetum*), while at high altitudes it assumes a primary role of permanent edapho-xerophilous vegetation. The best-preserved communities are located above the Timparossa beech forest (M. Nero, Etna N), above Mt. Conca (Piano Provenzano, Etna E), above Mt. Guardinazzi-M.Conca (Etna NW), above Mt. Palestra-Mt. Pecoraro (Etna W), between Mt. Scavo and Mt. Frumento Supino (Etna SW), and Mt. Nero degli Zappini (Etna SW). Making a comparison with the geological map of Branca et al. [22], it can be observed that the *Juniperus* areas are concentrated over, or near, the oldest emerging volcanic substrates of high altitude. This could mean that the oldest volcanic rocks (110–15 ka) functioned as refuge areas for the wood shrub species of the high-altitude volcanic environment [17]. In the Madonie Mountains, juniper is distributed at an altitude range between 1400/1500 and 1900 m a.s.l., mainly occupying the climatophilic series of the beech. On the Madonie, it was not possible to map the *Juniperus* vegetation due to the small and very fragmented patches [23]. The mountains affected by *Juniperus hemisphaerica*, especially on the very windy slopes facing south and south-west, are Monte Quacella, Monte Scalone, above Vallone Madonna degli Angeli, Monte Cavallo, and Monte San Salvatore. According to the reference grid (2 × 2 km), this habitat is currently recorded in six cells (45 geolocalized points) corresponding to a total surface area of 24 km^2^ (Figure 5). Our accurate field surveys allowed us to have a deeper knowledge of the distribution and conservation status of *Juniperus hemisphaerica* habitat. Based on our current assessments and observations in the field, the habitat is currently recorded in thirty cells (2 × 2 km), of which twenty-four cells are on Etna and six on Madonie (AOO = 120 km^2^) in Sicily. Considering the high phytogeographic value of the occurrence of many rare/endemic species in high-mountain juniper scrubs (*Rosa sicula* Tratt., *Rubus aetneus* Tornab., *Rosa heckeliana* Tratt., *Daphne oleoides* Schreb., *Genista cupanii* Guss., *Astragalus nebrodensis* (Guss.) Strobl, *Astragalus siculus* Biv., *Galium aetnicum* Biv., *Viola aethnensis* (Ging. & DC.) Strobl subsp*. aethnensis*, *Hieracium pallidum* Biv. subsp. *aetnense* Gottschl., Raimondo & Di Grist., *Bellardiochloa variegata* (Lam.) Kerguélen subsp. *aetnensis* (C.Presl) Giardina & Raimondo, *Allium nebrodense* Guss., *Jacobaea ambigua* (Biv.) Pelser & Veldkamp subsp. *nebrodensis* (Guss.) Peruzzi, N. G. Passal. & C. E. Jarvis, *Tanacetum vulgare* L. subsp. *siculum* (Guss.) Raimondo & Spadaro), and the remarkable naturalistic value of their habitat as well as their vulnerability to climate change, we propose their inclusion in the habitat 9560* “Endemic forest with *Juniperus* spp.” Annex I of the Habitat Directive (EUNIS habitat: F3.1a Lowland to montane temperate and submediterranean *Juniperus* scrub). Furthermore, on Etna, the dry juniper scrubs of high mountains are affected by bryophytic flora of high scientific and phytogeographic interest, such as *Rachytheciastrum collinum* (Schleich. ex Müll.Hal.) Ignatov & Huttunen, *Grimmia fuscolutea* Hook., *G. alpestris* (F. Weber & D. Mohr) Schleich., *Mielichhoferia elongata* (Hoppe & Hornsch. ex Hook.) Hornsch., and *M. mielichhoferiana* (Funck) Loeske [24]. Currently, the main threats to this habitat, in addition to the effects of climate change, are represented by grazing (sheep and goats) for both mountain systems (Madonie and Etna), and volcanic activities (ash and lava flows) for Etna only.

## 3. Materials and Methods

This study followed the Braun–Blanquet phytosociological approach [25]. A total of 156 phytosociological relevés × 151 species were collected, of which 52 were from the literature and 104 were unpublished. Based on the total set of phytosociological surveys, after eliminating the sporadic species with a presence lower than 4, a matrix of 156 surveys × 93 species was obtained. The original Braun–Blanquet sampling scale was transformed into the ordinal scale according to Van der Maarel [26], and transformation logx + 1 was applied in order to balance the highest and lowest cover values. All the relevés were analyzed using classification and ordination methods. A multivariate analysis (linkage method: Ward’s; distance measure: Euclidean) and detrended correspondence analysis (DCA) were applied [27,28,29]. Cluster analysis and ordination of the dataset were performed using PC-ORD 6 software [30]. The processed relevés from the literature were classified into the following syntaxa: *Cerastio tomentosi*–*Juniperetum hemisphaericae* Pignatti & Nimis [12]; *Lino punctati*–*Seslieretum siculae* subass. *juniperetosum* [13]; *Cerastio tomentosi*–*Juniperetum hemisphaericae* Pignatti & Nimis [13]; *Bellardiochloa aetnensis*–*Juniperetum hemisphaericae* Brullo & Siracusa [14]; and *Pruno cupanianae*–*Juniperetum hemisphaericae* Raimondo, Marino & Schicchi 2010 [15]. In addition, due to the high vegetation cover of *Juniperus hemisphaerica*, we also considered *Junipero hemisphaericae*–*Abietetum nebrodensis* Brullo & Giusso [14] and *Junipero hemisphaericae*–*Pinetum calabricae* Brullo & Siracusa [14]. A synoptic table (Table A2) was created to highlight the floristic composition, floristic richness, and altitude among the communities surveyed. Life form, chorology, and families were analyzed in a total of 151 vascular plant species extrapolated from 156 phytosociological relevés of the high- mountain juniper communities of Sicily. In order to define the altitudinal range and spatial distribution of *Juniperus hemisphaerica*, a total of 220 geolocated points for Etna and 45 points for Madonie were recorded, using GPS Garmin Montana and Quantum GIS software version 3.6. Taxonomic nomenclature, life form, and chorological classification of the endemics for native species follow Pignatti [31]. Syntaxonomic classification follows Mucina et al. [16]. Bioclimatic classification follows Pesaresi et al. [32]. The names of syntaxa comply with the International Code of Phytosociological Nomenclature (ICPN) [33].

## 4. Conclusions

This paper provides a comprehensive and exhaustive framework for the orophilous shrubby vegetation dominated by *Juniperus hemisphaerica* in Sicily, highlighting the structure, floristic composition, and dynamism of secondary and primary juniper communities. The results of this paper lead us to include the forests dominated by conifers of Sicily and southern Italy (*Junipero hemisphaericae*–*Abietetum nebrodensis*, *Junipero hemisphaericae*–*Pinetum calabricae*, *Junipero hemisphaericae*–*Abietetum apenninae*) in the alliance *Berberido aetnensis*–*Pinion laricionis* and order *Berberido creticae*–*Juniperetalia excelsae*. The dry juniper scrubs, due to their structure, floristic composition, and geographical distribution, are to be included in the new alliance *Berberido aetnensis*–*Juniperion hemisphericae*, proposed by us in this paper, and in the *Juniperetalia hemisphaericae* order, which best characterizes the juniper communities of Sicily from a geographical point of view.

The calcicolous supra-Mediterranean dry juniper scrubs (*Cerastio tomentosi*–*Juniperetum hemisphaericae* and *Pruno cupanianae*–*Juniperetum hemisphaericae*) of Mt. Madonie fall within the serial secondary aspects of *Luzulo siculae*–*Fagetum sylvaticae* and *Aceri campestri*–*Qurcetum ilicis*, respectively. Simultaneously, the silicicolous supra-oro-Mediterranean dry juniper scrubs (*Bellardiochloa aetnensis*–*Juniperetum hemisphaericae* and *Roso siculae*–*Juniperetum hemisphaericae*) of Mt. Etna fall into two different aspects: the first community plays a secondary serial role of the *Junipero hemisphaericae*–*Pinetum calabricae*, and the second community represents serial primary edapho-xerophilous vegetation of high mountains (permaseries). The endemism occurring in the high-mountain juniper communities is extremely high in both geographical contexts and is between 34% on Mt. Madonie and 43% on Mt. Etna, with a hemicryptophytic dominant biological form in both districts (56% for Etna and 60% for Madonie). The relict orophilous vegetation (permaseries), widespread only on Mt. Etna in an altitude range between 1900 m and 2300 m a.s.l., mainly on the northern and western rocky slopes, represents a habitat of extraordinary naturalistic value, not only due to the presence of several narrow endemic plants but also as an ecological niche for some faunal species, such as the Sicilian rock partridge *Alectoris graeca whitakeri* (Schiebel, 1834), now in strong decline, spotted several times in high altitude areas, where it finds shelter and food among the juniper scrubs. Moreover, this orophilous shrubby vegetation is very sensitive and vulnerable to climate change. It is also for these reasons that we propose, in this paper, the inclusion of this orophilous vegetation in the habitat 9560* “Endemic forest with *Juniperus* spp.” Annex I of the Habitat Directive.

## Figures and Tables

**Figure 1 plants-13-00423-f001:**
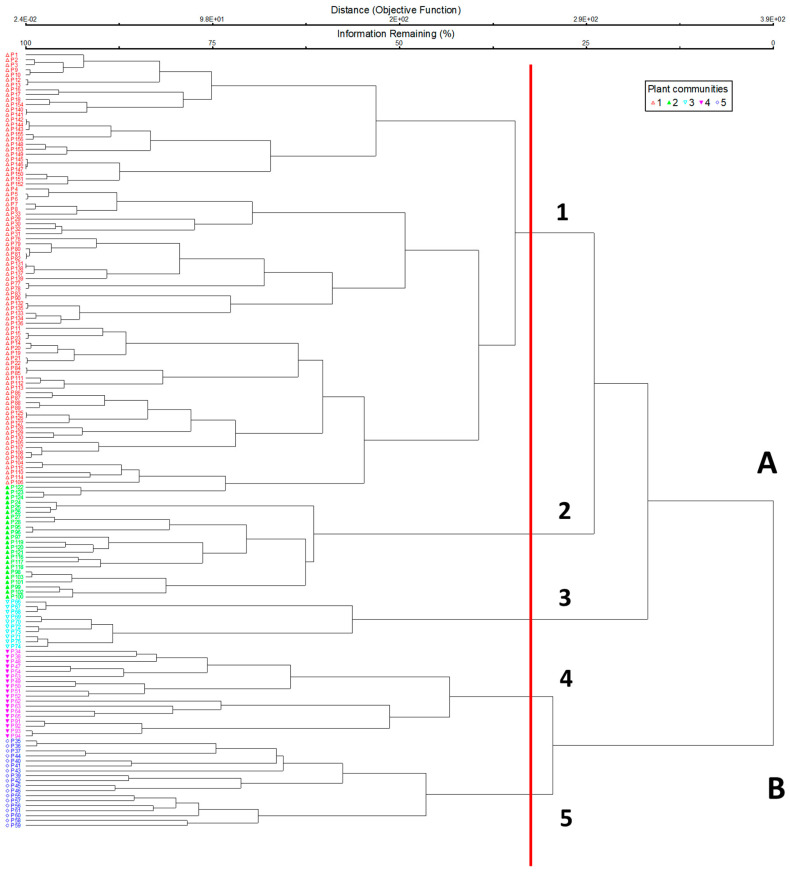
Cluster analysis of the plant communities ((**A**). Mt. Etna; (**B**). Mt. Madonie): 1. *Roso siculae*–*Juniperetum hemisphaericae*; 2. *Bellardiochloo aetnensis*–*Juniperetum hemisphaericae*; 3. *Junipero hemisphaericae*–*Pinetum calabricae*; 4. *Cerastio tomentosi*–*Juniperetum hemisphaericae* (including *Junipero hemisphaericae*–*Abietetum nebrodensis*); 5. *Pruno cupanianae*–*Juniperetum hemisphaericae* (including *Lino punctati*–*Seslieretum siculae subass. juniperetosum*).

**Figure 2 plants-13-00423-f002:**
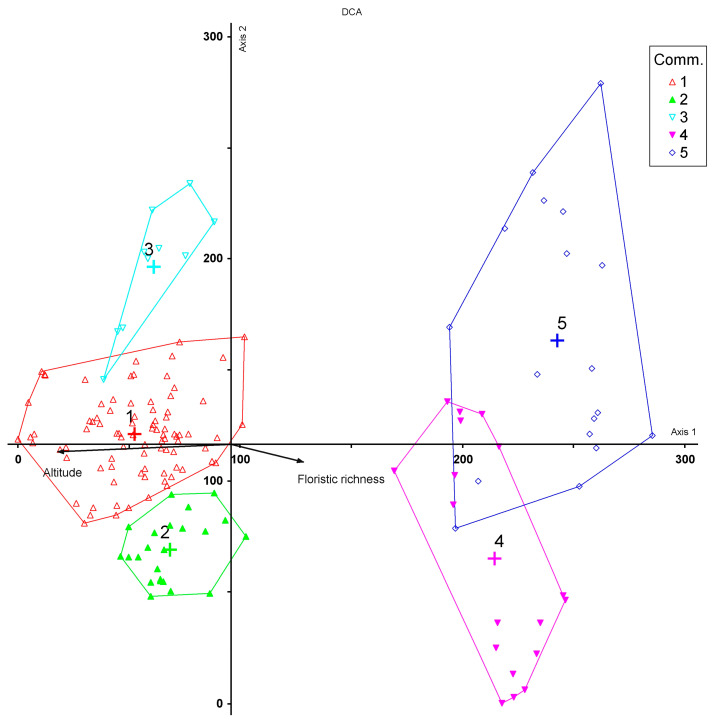
DCA of the plant communities. Total variance (‘inertia’) in the species data: 2.47. The r squared values of axes 1 and 2 are 0.39 and 0.21, respectively. Plant communities according to Figure 1.

**Figure 3 plants-13-00423-f003:**
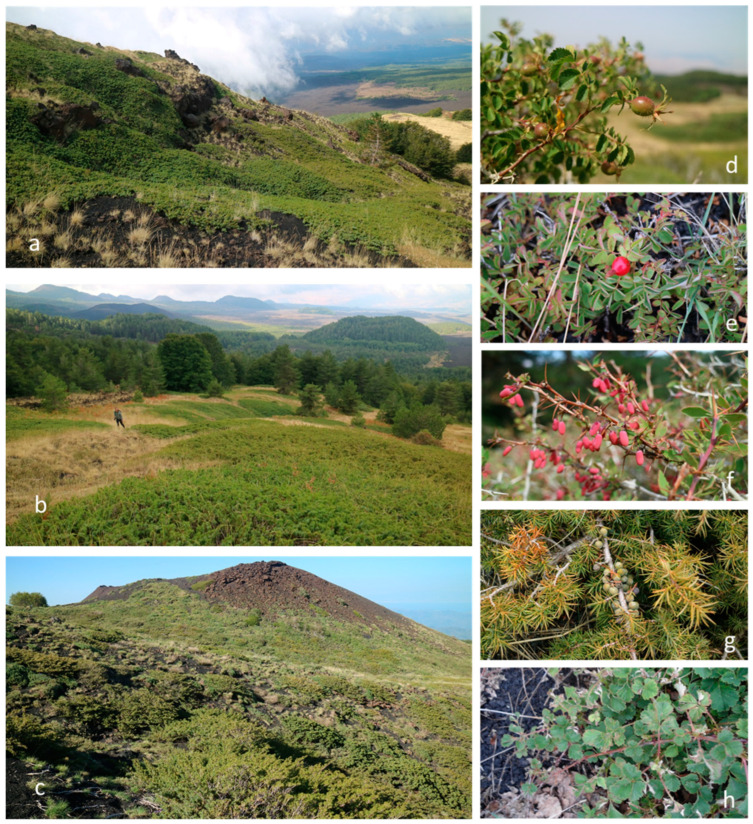
Photo plate illustrating the *Roso siculae*–*Juniperetum hemisphaericae* vegetation and diagnostic species of Mt. Etna: (**a**) Mt. Conca (NW); (**b**) Mt. Guardinazzi (NW); (**c**) Timparossa (N); (**d**) *Rosa heckeliana* Tratt.; (**e**) *Rosa sicula* Tratt.; (**f**) *Berberis aetnensis* C. Presl; (**g**) *Juniperus hemisphaerica* C. Presl; (**h**) *Rubus aetnicus* Cupani ex Weston.

**Figure 4 plants-13-00423-f004:**
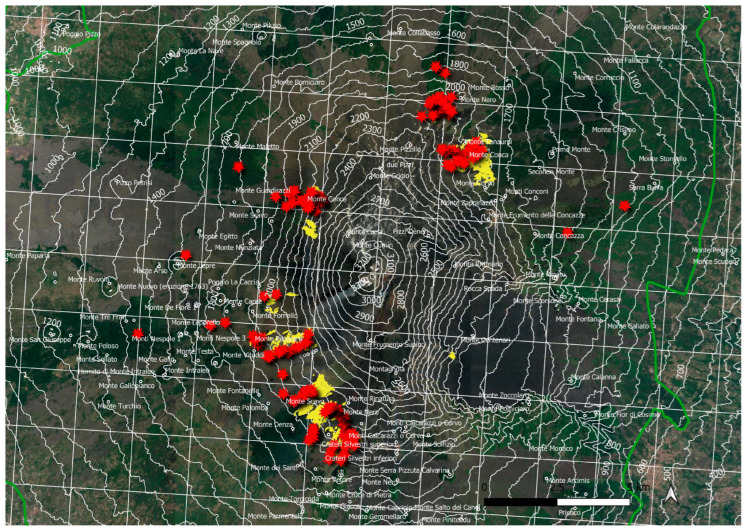
*Juniperus hemisphaerica* scrub distribution on Mt. Etna on a 2 × 2 km grid (yellow polygons and red stars).

**Figure 5 plants-13-00423-f005:**
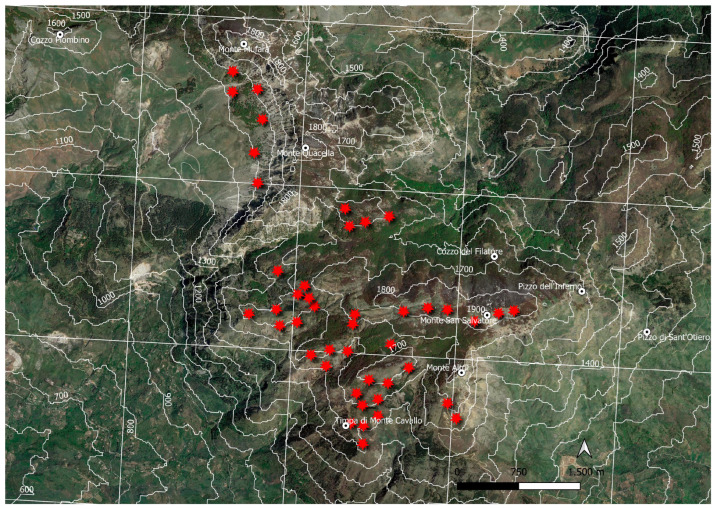
*Juniperus hemisphaerica* scrub distribution on Mt. Madonie on a 2 × 2 km grid (red stars).

## Data Availability

The data presented in this study are available on request from the corresponding author.

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
