# Peer review of "The Orophilous Shrubby Vegetation of the Juniperetalia hemisphaericae Order in Sicily: A Refuge Habitat for Many Endemic Vascular Species"

_plants, 2024, doi:10.3390/plants13030423_

Round 1

Reviewer 1 Report

Comments and Suggestions for Authors

Dear authors, please, consider the following remarks:

1.     The last paragraph of the Introduction chapter sounds like a “discussion”. The aim/s of the paper must be clearly stated there.

2.     Row 23: check if the word “survayes” is spelled correctly.

3.     The whole 2.1. Floristic data chapter looks somehow redundantly attached to the manuscript. Why is this there? It must be associated with the vegetation/syntaxa analyzed. Nothing about this chapter is mentioned in the Materials and Methods.

4.     In row 85 “A total of 151 vascular plant species”, but in row 381 “93 species”. Where this difference comes from? Should we consider that 58 species are present in less than 4 relevés, it is not clear? Don’t make readers guess.

5.     In row 91 “a.s.l.”, but in row 418 “slm”. Row 224: the same. Row 330: “above sea level”. Please, unify this throughout the entire manuscript.

6.     It is not clear why the new alliance proposed is considered a synonym of Berberidion aetnensis S. Brullo et al. 2001, when the latter is validated as Berberido aetnensis-Pinion laricionis (see rows 301, 302) and that means homotypic synonymy!?  So, what is the difference between the latter and the new Berberido aetnensis-Juniperion hemisphericae, when both appeared as synonyms of Berberidion aetnensis?

7.     A more detailed (floristic and ecological) description of the new alliance proposed is needed. Pointing the diagnostic species and the holotype only is not enough.

8.     Row 201: Ril.4? Is it relevé?

9.     Table 1 is missing! The absence of this table will cause an invalid publication of the new syntaxa proposed!

10.  Row 381: “156 phytosociological relevés per 93 species” doesn’t sound good, please, paraphrase.

Comments on the Quality of English Language

Some minor corrections needed. Please, see no. 2 and no. 10 listed in the Comments box. 

Author Response

Dear REV1

Thank you very much for the suggestions:

  1. The last paragraph of the Introduction chapter sounds like a “discussion”. The aim/s of the paper must be clearly stated there. Done
  2. Row 23: check if the word “survayes” is spelled correctly. Done
  3. The whole 2.1. Floristic datachapter looks somehow redundantly attached to the manuscript. Why is this there? It must be associated with the vegetation/syntaxa analyzed. Nothing about this chapter is mentioned in the Materials and Methods. Done
  4. In row 85 “A total of 151 vascular plant species”, but in row 381 “93 species”. Where this difference comes from? Should we consider that 58 species are present in less than 4 relevés, it is not clear? Don’t make readers guess. Done
  5. In row 91 “a.s.l.”, but in row 418 “slm”. Row 224: the same. Row 330: “above sea level”. Please, unify this throughout the entire manuscript. Done
  6. It is not clear why the new alliance proposed is considered a synonym of Berberidion aetnensisS. Brullo et al. 2001, when the latter is validated as Berberido aetnensis-Pinion laricionis (see rows 301, 302) and that means homotypic synonymy!?  So, what is the difference between the latter and the new Berberido aetnensis-Juniperion hemisphericae, when both appeared as synonyms of Berberidion aetnensis? it's a typo. Done
  7. A more detailed (floristic and ecological) description of the new alliance proposed is needed. Pointing the diagnostic species and the holotype only is not enough. Done
  8. Row 201: Ril.4? Is it relevé? Done
  9. Table 1 is missing! The absence of this table will cause an invalid publication of the new syntaxa proposed! Table1 was attached separately, I don't understand why reviewers can't see it.
  10. Row 381: “156 phytosociological relevés per 93 species” doesn’t sound good, please, paraphrase. Done

Reviewer 2 Report

Comments and Suggestions for Authors

Dear Authors,

I want to begin by pointing out that I am somewhat disappointed with the manuscript presented. I believe that formations like quasi-forests in high mountains dominated by conifers are extraordinarily interesting and highly threatened by climate change and global environmental shifts. By the way, not a word is mentioned throughout the manuscript about this.

I am also disappointed because I am familiar with works already published by the authors, which I consider highly interesting. However, throughout the manuscript, I have the impression that this document has been prepared somewhat hastily. And this is not the only problem I find; the proposals made regarding the communities or associations suggested are not supported by the data.

I think, to a large extent, this discrepancy between the analyses and the results offered is due to the fact that the data have not been properly treated. With the sincere intention of improving the manuscript, I will allow myself to make some suggestions.

The first one is to try to adjust the title, abstract, introduction, ... in general, all sections of what constitutes a scientific article. For example, since the article does not focus on threatened species, why include "refuge for threatened plants" in the title?

On the other hand, in a syntaxonomic study like this, a synthetic table summarizing floristic differences between associations is missing. These differences should be supported by diagnostic species of the Pino-Juniperetea class, perhaps accompanied by other differential species from other syntaxonomic units important in high mountain areas. If Iberian associations mentioned are included in this table, the research would have a broader spectrum and gain a larger audience. Please seriously consider this recommendation.

Additionally, review the multivariate analyses. Besides the transformation using the van der Maarel scale, it is likely that the data should be transformed to logx+1 or square root. This would reduce the influence of highly abundant species and give prominence to others. Keep in mind that syntaxonomy is more qualitative than quantitative. With the Ward method, it is better, almost obligatory, to use Euclidean distance.

Other comments are available directly on the manuscript.

I hope these suggestions contribute to the improvement of your manuscript.

Author Response

Dear Rev2

thanks for all the suggestions that enriched the text

I want to begin by pointing out that I am somewhat disappointed with the manuscript presented. I believe that formations like quasi-forests in high mountains dominated by conifers are extraordinarily interesting and highly threatened by climate change and global environmental shifts. By the way, not a word is mentioned throughout the manuscript about this. I'm very sorry about this, but the main objective of our manuscript is the syntaxonomic aspect and to a small extent also floristic. In reality, there is a study underway, which will require further years of monitoring, which concerns the migrations of Etna species (bottom and top) influenced by climate change.

I am also disappointed because I am familiar with works already published by the authors, which I consider highly interesting. However, throughout the manuscript, I have the impression that this document has been prepared somewhat hastily. And this is not the only problem I find; the proposals made regarding the communities or associations suggested are not supported by the data. I am very sorry to contradict this thought, but our manuscript is supported by several unpublished phytosociological releves, collected with criterion, method and rigor (made with a lot of effort) see table 1, in addition to the statistical calculations. On a mountain/volcano like Etna it is not easy to reach some places (only on foot) to survey Juniperus vegetation in a primary environment (Permaseries). No one had ever climbed so high up Etna to survey these well-structured aspects, rich in shrub species.

I think, to a large extent, this discrepancy between the analyses and the results offered is due to the fact that the data have not been properly treated. With the sincere intention of improving the manuscript, I will allow myself to make some suggestions.

The first one is to try to adjust the title, abstract, introduction, ... in general, all sections of what constitutes a scientific article. For example, since the article does not focus on threatened species, why include "refuge for threatened plants" in the title? I would like to point out that the title refers to endemic and non-threatened species, which are two very different things. I would like to leave this title because the manuscript, even if to a lesser extent, also deals with the overall floristic aspect of the juniper groves, in particular endemic species and biological forms, between the two highest mountains of Sicily, which I find very interesting also to stimulate further floristic studies on climate changes. But if you don't agree I'm also willing to eliminate part of the title.

On the other hand, in a syntaxonomic study like this, a synthetic table summarizing floristic differences between associations is missing. These differences should be supported by diagnostic species of the Pino-Juniperetea class, perhaps accompanied by other differential species from other syntaxonomic units important in high mountain areas. If Iberian associations mentioned are included in this table, the research would have a broader spectrum and gain a larger audience. Please seriously consider this recommendation. The idea is very interesting, I tried to insert a table in the text that summarizes the communities of Sicily. It's true it would be interesting to also include the Spanish ones which would take me more time.

Additionally, review the multivariate analyses. Besides the transformation using the van der Maarel scale, it is likely that the data should be transformed to logx+1 or square root. This would reduce the influence of highly abundant species and give prominence to others. Keep in mind that syntaxonomy is more qualitative than quantitative. With the Ward method, it is better, almost obligatory, to use Euclidean distance. Really thanks for these suggestions, even with Euclidean distance it changes little, as you can see in the cluster below (see text attac).

Furthermore, almost all the suggestions indicated in the pdf have been included in the text

Round 2

Reviewer 1 Report

Comments and Suggestions for Authors

Dear authors,

Thank you for taking the time to revise your manuscript. I am satisfied with the new version. The revised manuscript is much improved.

Author Response

Dear REV

Thank you so much for your valuable suggestions

SS

Reviewer 2 Report

Comments and Suggestions for Authors

Dear authors,

I regret that my comments have been of so little help in improving your manuscript. In this new revision, I urge you to focus on improving the cluster. It is very possible that I have misunderstood something, but if the samples highlighted in red with the triangle symbol refer to the Rosa sicula and Juniperus community, there are about 20 relevés that fall within the Bellardiochlo cluster. Therefore: 1.- either they are all from the same association (Roso-Juniperetum), or 2.- the cluster marked as two is a simple association.

In any case, the cladogram is confusing since the colors and symbols used do not correspond to the groups 1-5 employed. The impression conveyed is that there is an attempt to see what one would like to see rather than reflecting reality. For example, it is impossible to consider a different association for the set of 4 relevés reflected in the cluster analysis as Junipero-Abietetum. What is the criterion for this?

But perhaps more importantly, the number of samples or relevés in the cluster and PCA does not match. What is the reason for this discrepancy? For instance, where is that group of 4 inventories of Junipero-Abietetum in the PCA? To make matters worse, in the cluster, there are 6 symbols (+colors) and only 5 groups. One of these groups, present in the previous version of the manuscript in the PCA, has disappeared in this new version.

I would like to insist that for figures 1 and 2, please use the same colors and symbols. Otherwise, it will confuse the readers, and they won't understand anything. At least, that's the case for me. Keep in mind that there may be researchers interested in your work who believe that the colors and symbols used for both types of analysis conducted with the same data should be consistent. There are very meticulous people out there.

In conclusion, the two figures upon which the analysis of the results is based are very confusing, discordant, and do not confirm the conclusions; instead, they cast doubt upon them. I believe you should fix this if you want your manuscript to be more comprehensible and rigorous.

I think the synoptic table is helpful, but I don't understand why there are 4 associations in the syntaxonomic scheme and 6 in that table.

Believe me, I am trying to help with these comments, but from this point onward, I'll let you continue with your manuscript without the burden of my comments.

Thank you very much for taking some of them into account, and good luck with your research.

Author Response

Dear 

Thank you very much for your very useful suggestions. I know well that these are just revisions made to improve the manuscript and me too. I tried to follow your advice and redo both the cluster and the DCA, using the Ward method with the Euclidean distance and also the logarithmic transformation. The result has significantly improved. I also standardized Cluster, DCA and Table2. Regarding the Junipero hemisphaericae-Abietetum nebrodensis is considered in the cluster due to the high values of Juniperus hemisphaerica and low values of Abies nebrodensis. This is indicated in the text notes. In fact, in the cluster it is included in the Cerastio-Jumiperetum. For this reason it was included in the DCA in Cerastio-Juniperetum.

I hope now that you like the manuscript at least a little.

tks for all

SS

Round 3

Reviewer 2 Report

Comments and Suggestions for Authors

I think the manuscript is now clearer and the conclusions are supported by the data. Remember that some of the suggestions I have made could help you continue improving your knowledge of these interesting high mountain communities.

Luck for the future